# MariClus: Your One-Stop Platform for Information on Marine Natural Products, Their Gene Clusters and Producing Organisms

**DOI:** 10.3390/md21080449

**Published:** 2023-08-15

**Authors:** Cedric Hermans, Maarten Lieven De Mol, Marieke Mispelaere, Anne-Sofie De Rop, Jeltien Rombaut, Tesneem Nusayr, Rebecca Creamer, Sofie L. De Maeseneire, Wim K. Soetaert, Paco Hulpiau

**Affiliations:** 1Bioinformatics Knowledge Center (BiKC), Campus Brugge Station, Howest University of Applied Sciences, Rijselstraat 5, 8200 Bruges, Belgium; cedric.hermans2@howest.be (C.H.);; 2Centre for Industrial Biotechnology and Biocatalysis (InBio.be), Department of Biotechnology, Faculty of Bioscience Engineering, Ghent University, Coupure Links 653, 9000 Ghent, Belgium; 3Life Sciences, Texas A&M-Corpus Christi, Corpus Christi, TX 78412, USA; 4Entomology, Plant Pathology, and Weed Science, New Mexico State University, Las Cruces, NM 88003, USA

**Keywords:** MariClus database, marine prokaryotes, genome mining, gene clusters, natural product discovery

## Abstract

Background: The marine environment hosts the vast majority of living species and marine microbes that produce natural products with great potential in providing lead compounds for drug development. With over 70% of Earth’s surface covered in water and the high interaction rate associated with liquid environments, this has resulted in many marine natural product discoveries. Our improved understanding of the biosynthesis of these molecules, encoded by gene clusters, along with increased genomic information will aid us in uncovering even more novel compounds. Results: We introduce MariClus (https://www.mariclus.com), an online user-friendly platform for mining and visualizing marine gene clusters. The first version contains information on clusters and the predicted molecules for over 500 marine-related prokaryotes. The user-friendly interface allows scientists to easily search by species, cluster type or molecule and visualize the information in table format or graphical representation. Conclusions: This new online portal simplifies the exploration and comparison of gene clusters in marine species for scientists and assists in characterizing the bioactive molecules they produce. MariClus integrates data from public sources, like GenBank, MIBiG and PubChem, with genome mining results from antiSMASH. This allows users to access and analyze various aspects of marine natural product biosynthesis and diversity.

## 1. Introduction

Marine life has been around for about 3.7 billion years, which is much longer than terrestrial life. Not only is there a greater evolutionary diversity in marine environments but the vast majority of marine species are undescribed. This offers a wealth of opportunities for scientific research that can result in innovative solutions to the various challenges we face today, such as antimicrobial resistance, viral pandemics and increasing cancer prevalence. Marine bacteria are naturally wired with a primary and secondary metabolism comprising various biochemical reactions to support cellular processes, enabling them to live in a highly competitive environment [1]. While the primary metabolism produces molecules that derive energy and enable growth, the secondary metabolism plays an important ecological function in communication and competition by manufacturing specialty compounds. These unique substances are encoded by genes in the genome, which are typically clustered together in so-called gene clusters. Examples of such specialty compounds produced by marine species are as follows: polyketides, nonribosomal peptides (NRPs), ribosomally synthesized and post-translationally modified peptides (RiPPs), terpenes and alkaloids [2,3,4]. While it is hard to determine an exact number of drugs derived from natural products, it is safe to say that at least 50% of drugs that have been approved for clinical use are derived from substances identified from natural sources [5]. In particular, marine natural products have demonstrated their potency in the past, and an upward trend in approved drugs has been noted [6,7,8]. Although the list of drugs in clinical use derived from marine species is still limited, previously described and recently discovered substances show great promise, leading to an increased interest in microbial drug discovery of species inhabiting marine environments [9,10,11].

To facilitate natural product discovery, various databases and online tools, such as PubChem and antiSMASH, have been developed. Specifically developed for marine research, the Marine Metagenomics Portal (MMP) [12], Comprehensive Marine Natural Products Database (CMNPD) [13] and MarinLit [14] are available. MMP offers a comprehensive overview of discovered marine species, including metadata on their environment and location, as well as providing several services to analyze the genomes of these species. However, these analyses are rather exploratory and do not enable marine natural product discovery in a transparent manner. Information pertaining to substances discovered in marine environments is manually curated in the CMNPD, which provides access to their source, cellular targets, bioactivities, patents and scientific publications [13]. The latter is also covered by the MarinLit database, hosted by the Royal Society of Chemistry, which requires a paid subscription for access. Such databases offer centralized information that aids in explaining the complexity of marine environments and the biochemical synthesis routes for natural substances produced by its residents.

The gold standard and most widely used tool for mining bacterial genomes and discovering gene clusters today is antiSMASH [15]. It is available as a standalone tool, for which some bioinformatics expertise is required, or a genomic sequence can be analyzed in the open-source web server. The development of next-generation and, more recently, third-generation sequencing technologies, along with a decrease in sequencing costs, has made it possible to obtain a vast number of genomes. Analyzing such a large amount of genomic data presents a challenge. The NCBI database [16] currently holds more than 1.5 million bacterial genomes. These are unfortunately not all of the same quality as, for example, the assembly might be incomplete, meaning that the genome consists of many contigs, which could render unreliable genome mining results. Furthermore, for some species, many different assemblies are available. Fortunately, for most species with multiple assemblies, NCBI designates one assembly as either ‘reference’ or ‘representative’. This collection of ‘representative’ and ‘reference’ genomes provides a concise, standardized and taxonomically diverse view of the genome collection. To facilitate research on natural product biosynthesis, we developed MariClus (https://www.mariclus.com), a platform that combines genomics and bioinformatics results. At the time of writing, over 500 high-quality marine prokaryote genomes from NCBI datasets have been analyzed using antiSMASH 7. The discovered gene clusters and other metadata have been uploaded to this online portal, which is freely available to the scientific community.

## 2. Results

### 2.1. Data Collection

The initial list of species prior to data retrieval was collected and curated by members of the different research groups co-authoring this article. To include marine species, the criteria were as follows: availability of a high-quality genome assembly in NCBI Genome datasets, and the (main) source of the species needed to be a marine environment. For example, the genus *Salinispora*, which has been proven to be a rich source of unusual secondary metabolites, had, at the time of writing, eight species with genome entries: *S. arenicola*, *S. pacifica*, *S. tropica*, *S. fenicalii*, *S. vitiensis*, *S. oceanensis*, *S. mooreana* and *S. cortesiana*. Their NCBI Genome records were investigated, and the representative or reference genome accession number (GCF/GCA) was retrieved together with other information such as isolation source from the NCBI Genome or GenBank record. Additionally, the BacDive [17] record or relevant publications were investigated depending on the availability of the isolation source information. In this case, *S. tropica* had two genome assemblies to choose from. The first strain CNT250 with ‘scaffold’ assembly level consisted of 59 scaffolds. The second strain CNB-440 had assembly status ‘complete’, consisting of only one scaffold (GCF_000016425.1) and was the representative genome selected for genome mining and inclusion in MariClus. Both the study of [18] and BacDive reported a marine origin for *S. tropica* isolated from marine sediment. Using this methodology, we manually collected and curated species on their assembly level, taxonomy, isolation source and relevance and collected genomic data for more than 500 marine species for further analysis. The high-quality genome assemblies were analyzed using antiSMASH 7 to identify gene clusters and predict their products. We also extracted additional information about the species, such as relevant publications and culture collection number, from various sources, such as PubMed and NCBI Taxonomy. We then integrated all the data into a MariaDB database and developed a web interface to display and query the data.

### 2.2. Online Portal

The MariClus online portal was designed to be user-friendly and intuitive, allowing users to easily access and explore the data on marine gene clusters and molecules. The portal consists of four main data tabs: Dashboard, Species, Clusters and Molecules. Each tab provides different functionalities and information for users.

The Dashboard tab offers a summary of the data in the database, such as the number of species, clusters and molecules, as well as some statistics on the database content, such as the distribution of cluster and molecule types. It also provides direct links to the Species, Clusters and Molecules pages where users can search and filter the data on demand.

The Species tab allows users to search for marine species by name, NCBI genome assembly ID, PubMed ID or (manually curated) additional information on the species. By clicking on the details icon or species name, users can view more information about the species, such as its taxonomic lineage, genome assembly, relevant publications and culture collection number. Furthermore, a list of clusters and molecules predicted for the species, and access to their respective antiSMASH pages by clicking on the icons, is given.

The Clusters tab allows users to search for gene clusters by their type or contig. Users can also filter the results by the length of the cluster or number of genes associated with each cluster. By clicking on the details icon, the user is forwarded to the same page as they would for the species. The page will automatically scroll to, and highlight, the cluster of interest.

The Molecules tab lets users easily search the database for specific molecules that are predicted by antiSMASH. It also allows you to see the contig and cluster type as well as the cumulative BLAST score, which was calculated by the KnownClusterBlast in antiSMASH. A higher cumulative BLAST score indicates that more/larger hits have been found with a known gene cluster entry in the Minimum Information about a Biosynthetic Gene cluster (MIBiG) database, which is an online repository containing curated clusters with known products.

In addition to these main tabs, the website also hosts a general search bar, which lets you search through the entire database at once. For convenience, the results are split into three sections. The first section focuses on the species, including the description and NCBI accession number for the assembly, as well as any related PubMed articles stored for the species. The second section focuses on the gene clusters and molecules while also showing the contigs on which the clusters are located. The third and final section on this page focuses on the PubMed articles in the database and lets you search through the titles of these articles. The search term for every result is also highlighted in the table to show the user where the match occurred. All of these sections link to the detailed species page, which contains all information for this species. An example search of ‘salinosporamide’ is shown in Figure 1. Clicking on *Salinispora tropica* within the ‘Gene clusters and molecules’ section brings you to the details page of this species where predicted products are collapsed, with the exception of the salinosporamide query hit ‘salinosporamide A’ (Appendix A).

To facilitate interaction with the scientific community and MariClus users, the platform contains a contact page to report bugs, correct inconsistencies or request specific assemblies to be added to the database.

Finally, a tutorial page is available to help users with how the platform should be used to search for and explore the data on marine species, gene clusters and molecules. The tutorial page contains screenshots and explanations of the different features and functionalities of the website, as well as some examples of queries and results. It is intended to help users become familiar with the portal and make the most of its capabilities.

### 2.3. Meta-Analyses

To provide a comprehensive overview of the data, several meta-analyses were performed. These analyses include a taxonomic analysis, the number of clusters per cluster type and the number of clusters per molecule. On average, every assembly has 5.83 clusters, with a standard deviation of 6.02 clusters. Of the 545 species that were analyzed, 37 did not have any cluster detected by antiSMASH. As various predictions are often associated with one cluster, the number of predicted molecules is higher than the number of clusters. On average, the number of molecules per assembly is 41.24, with a standard deviation of 95.54 molecules.

The assembly with the most detected clusters is GCA_001854205.1, which is an assembly corresponding to the species *Moorena producens*. This species is a benthic filamentous cyanobacterium that has a global distribution and can form harmful blooms in marine ecosystems. It produces a variety of bioactive secondary metabolites, such as malyngamides, microcolins and dolastatins, that have toxic effects on various organisms and human health but also potential pharmaceutical applications [19]. Despite having the highest number of detected clusters, it does not have the most predicted molecules. The assembly with the most predicted molecules is GCF_008704715.1, which is an assembly corresponding to the species *Streptomyces chartreusis*.

#### 2.3.1. Taxonomical Statistics

The current version of the database contains over 500 prokaryote species. Most of these (88%) belong to the superkingdom bacteria; however, there are also some species that belong to the superkingdom archaea. For bacteria, the most common (super)phyla are Pseudomonadota, the Fibrobacterota, Chlorobiota and Bacteroidota (FCB) group and Terrabacteria. Pseudomonadota and Terrabacteria are also the most common (super)phyla in the NCBI database, totaling, at the time of writing, over 80% of all bacterial species (Appendix A) [20]. The FCB group is a very common group of bacteria in marine environments, especially in marine sediment [21,22]. For archaea, the most common (super)phyla are Euryarchaeota and the Thaumarchaeota (now Nitrososphaerota), Aigarchaeota, Crenarchaeota (now Thermoproteota) and Korarchaeota (TACK) group. Both Euryarchaeota and the TACK group are also the most common (super)phyla in the NCBI database. Less common, but still present in decent numbers, is the Thermodesulfobacteriota group, which is a group of bacteria that are often associated with hydrothermal vents, thermophilic digestors, hot springs and other high-temperature environments [23]. The (super)phyla with over 10 species in the database are shown in Figure 2A.

#### 2.3.2. Cluster Types

The database contains almost 3000 clusters, which are groups of genes that work together to produce a specific molecule. These clusters are categorized into 65 different types, with the most common ones being terpene, nonribosomal peptide synthetase (NRPS), RiPP-like, type-1 polyketide synthase (T1PKS), betalactone and NRPS-like (Figure 2B). Terpenes are the most common type in the database, with over 500 clusters. NRPS clusters come in second with over 200 clusters. RiPP-like clusters are close to 200 clusters, and the T1PKS, betalactone and NRPS-like types are just over 150 clusters. All these types of molecules are very well known within marine bacteria and are often the subject of interest for studies regarding novel antibiotic and anticancer agents or to combat various other diseases [2,24,25]. An overview of the top 10 most common types of clusters is shown in Figure 2B.

#### 2.3.3. Molecule Types

The antiSMASH 7 software predicts the molecules that are produced by the clusters using the module KnownClusterBlast. This module compares the clusters to curated clusters in the MIBiG database and uses this information to predict compounds possibly produced by the clusters. Often, antiSMASH will have multiple predictions for a single cluster as these clusters can share extensive similarities between different compounds. The database contains over 16,000 molecules that are predicted to be produced by the clusters. The total number of unique molecules is over 1200. The most common molecules are ectoine and carotenoids. Ectoine is the most prominent osmolyte in nature, and it is produced by many bacteria and archaea to protect them from multiple kinds of stress [26,27]. Carotenoids are the most common group of pigments that are produced by many organisms, including marine bacteria. They are best known for their antioxidant function during photosynthesis [28].

### 2.4. Case Studies

To showcase the functionality and user-friendliness of MariClus, several case studies pertaining to different scientific fields were tested and are given below.

#### 2.4.1. Microbiology

With over 500 marine bacterial species, MariClus holds huge potential for marine research pertaining to marine biodiversity and environmental habitats. The search field under the ‘Species’ tab allows the user to quickly gather reliable information on bacterial species identified in various geographic locations. For example, by searching for ‘China’, 25 bacterial species were retrieved with data on their isolation source (e.g., marine sediment, seawater or deep sea), often their general location (e.g., Zhoushan Islands, South China Sea or coast of Weihai), as well as links to relevant publications. Details on the strains can be obtained by clicking on the investigation loop symbol, linking other database functionalities, such as gene clusters uncovered with antiSMASH or potential marine molecules associated with these gene clusters. It also allows the user to check whether the investigated strain is available in culture collections and can be ordered for further characterization. For example, *Aequorivita iocasae*, one of the strains isolated from Chinese waters, has a BacDive profile and is purchasable from the Japanese Collection of Microorganisms (JCM) under the culture collection number JCM 34635. Should this strain require further phenotypic, genomic or marine natural product investigation, information is available on MariClus.

To test the MariClus resource for bacterial clusters associated with bacteria that degrade oil in cold marine environments, a search was run using the term ‘North Sea’. The first search gave five bacterial species from very different environments. One of these, *Alcanivorax borkumensis*, is a ubiquitous hydrocarbon-degrading marine bacteria. A search for ‘Antarctic’ gave 13 hits, including *Oleispira antarctica*, another ubiquitous oil degrader. Interestingly, searching for hydrocarbon degradation provides two additional bacteria. MariClus, therefore, was helpful in identifying the bacteria from cold deep-sea environments with the ability to degrade oil.

A search for ‘Hydrothermal vent’ in the general search function on top of the Dashboard identified 21 different species and five articles. The description for these 21 species mentions from where these species were isolated. For example, *Aciduliprofundum boonei* was isolated from the mariner hydrothermal vent field in the South Pacific Ocean, *Gimesia algae* was isolated from alga collected from the hydrothermal vent system and for *Hydrogenovibrio crunogenus*, the description mentions it is a chemoautotroph commonly isolated from deep-sea and shallow-water hydrothermal vents. Furthermore, for *H. crunogenus,* the description says ‘This organism uses the oxidation of reduced sulfur compounds (thiosulfate, sulfide and sulfur) to generate the energy necessary to fix carbon and plays an important role in the cycling of sulfur in the marine environment’.

As demonstrated, the general search function on the top of the page can help find bacteria that match a keyword. However, using multiple words may limit the results, as the search function only looks for exact matches. For instance, the keyword ‘sulfur’ returns 15 bacteria that use sulfur in their metabolism, such as *Hydrogenovibrio crunogenus*. But, adding another word, such as ‘sulfur compounds’, reduces the results to only two bacteria, one of them still being *H. crunogenus*. A similar pattern occurs with the keyword ‘*Arcobacter* sp.’, which shows 8 different species that are resistant to antibiotics and 21 species under Molecules. But, a more specific keyword, such as ‘antibiotic resistance’, shows only one article referring to one bacterium, *Aliarcobacter butzleri*. This keyword misses *Marivirga tractuosa*, a bacterium that is also resistant to antibiotics and has a complete genome sequence available. The reason is that the database contains the text ‘resistance to several antibiotics’ for this bacterium, which does not match the keyword ‘antibiotic resistance’. However, using a simpler keyword, such as ‘resistance’, shows both *Aliarcobacter butzleri* and *Marivirga tractuosa*, as well as two articles. Thus, using one keyword can increase the chances of finding more relevant bacteria with the general search function at the top of the page.

#### 2.4.2. Gene Cluster Identification

MariClus also contains, next to information on marine species, data on their genome architecture and coding elements. While (new) natural products are continuously being reported in the literature, the coding sequences or enzymes responsible for these compounds are often not investigated or are difficult to uncover [4,29,30]. Unearthing the biochemical pathways leading to marine natural products heightens the fundamental understanding of how and when these compounds are synthesized, potentially steering research towards new discoveries or the expression of silent gene clusters. In addition to these fundamental insights, the knowledge on genes and enzymes makes it possible to construct microbial cell factories by transferring genes to a different, heterologous host, thereby providing an alternative to chemical synthesis or extraction production processes. Should one be interested in, for example, the microbial production of the polyketide antifolate molecule abyssomicin, one can search for ‘abyssomicin’ under the ‘Molecules’ section of MariClus. This will retrieve a list of marine species and gene clusters associated with abyssomicin. Abyssomicin is a unique antimicrobial as eukaryotes do not natively synthesize folate [31]. This list is standardly ranked according to a cumulative BLAST score of the genes present in the identified cluster compared to the known abyssomicin C gene cluster of the *M. maris* AB-18-032 isolate to ensure that the most likely hit is given on top, which is demonstrated, in this case, as *Micromonospora maris* is ranked highest. By clicking on the details icon, the user is taken to the respective gene cluster, which is highlighted in yellow. As information on public databases and predictions with genome mining tools such as antiSMASH are not always accurate, MariClus allows the user to make their own conclusion by providing details on the antiSMASH product predictions, accessed by clicking on ‘Evidence’. In the case of the abyssomicin gene cluster of *M. maris*, 27 out of 28 genes from the cluster were correlated to the known abyssomicin C gene cluster available on the MIBiG database with accession number BGC0000001. The antiSMASH icon given with the gene cluster will bring the user to the antiSMASH gene cluster webpage, which contains all features normally available when using antiSMASH as a standalone tool. While the KnownClusterBlast on the antiSMASH webpage (Figure 3) enables graphic examination of the candidate gene cluster from *M. maris* with known gene clusters, it is, by default, limited to 10 hits and does not provide numerical evidence of the BLAST results. The ‘Evidence’ information of gene clusters on MariClus elaborates on the antiSMASH results by displaying up to 40 additional hits and, more importantly, offers the BLAST values for each gene in the cluster. In this way, MariClus provides a comprehensive and transparent report on the antiSMASH natural product prediction by gene clusters.

MariClus not only provides transparent information on gene clusters within a species but also displays similar clusters from other marine bacteria. Next to *M. maris*, *Streptomyces buecherae*, *Saccharomonospora piscinae*, *Streptomyces marincola*, *Actinoalloteichus fjoriducs* and *Amycolatopsis albispora* are ranked high when searching for abyssomicin. From Table 1, it is clear that the cumulative BLAST score is highest for *M. maris*, as expected, but that the gene cluster of the abyssomicin C gene cluster from *M. maris* AB-18-032 possesses elements in an architecture similar to gene clusters of various other natural products. Despite the high BLAST score of the gene cluster of *S. buecherae* for abyssomicin, the best hit is actually for the nigericin gene cluster of *Streptomyces violaceusniger* (accession number BGC0000114 on MIBiG). Contrary to the significant resemblance of the *S. buecherae* gene cluster with the nigericin gene cluster, the likelihood that the gene clusters of *S. piscinae* and *A. fjordicus* are associated with abyssomicin or mediomycin A is rather slim, as only part of the gene clusters is associated with the known clusters, reflected by the coverage in Table 1. The best hits for *S. marincola* and *A. albispora* are predicted to be the heronamide and quartromicin clusters, respectively (Figure 3), which matches perfectly with the known BGC clusters for these molecules [33,34,35]. To conclude, the MariClus repository is capable of retrieving the gene cluster of interest and allows for an in-depth analysis of similar gene clusters that might provide orthologous enzymes for enzyme characterization or metabolic engineering. It makes use of the data generated by antiSMASH 7, presents it in a user-friendly manner and enables enhanced genome mining as it is not limited to comparing a certain gene cluster to only 10 known ones (as is the case if antiSMASH was used as a standalone tool).

#### 2.4.3. Natural Product Discovery

As marine natural products are a valuable source of pharmaceutical drugs, they have been widely researched [8]. MariClus, therefore, links the predicted compounds produced by marine species to their PubChem page where additional information concerning biochemical pathways, assays, studies and patents can be found. For example, the marine natural product salinosporamide A was discovered in 2003 from marine *Salinispora* species and has since been commercialized as the proteasome-inhibiting anticancer drug Marizomib [36,37]. Consequently, many papers and patent applications have been published. The PubChem icon next to salinosporamide A of *Salinispora tropica* in the MariClus database directly links to PubChem where one can find an overview of 89 substances, 21 related compounds (generalized information of multiple substances), 3 biochemical pathways, 137 assays, 103 manuscripts and 100 patents.

## 3. Discussion

Despite the historical success of natural products as medical treatments, challenges in their discovery and technical issues have led to a decline in interest from pharmaceutical companies in recent decades. However, the fight against antimicrobial resistance and recent technological and scientific advances are reviving this interest [38]. Genome mining and increasing knowledge on biochemical pathways of natural products are assisting modern drug discovery. AntiSMASH is the most widely used tool for the identification of genes clustered together in the genomes of bacteria and fungi [15]. Such predictive tools provide hints about the novelty of the metabolic products of clusters and, when combined with spectroscopic techniques, can accelerate the identification of valuable natural products. To easily extend genome mining from a single genome to entire collections of bacterial genomes for exploration and comparative cluster analysis, several requirements must be met: genome assemblies need to be of sufficient quality, i.e., not too fragmented, antiSMASH must be run in batch mode with sufficient computing time and data storage, depending on the amount of genome assemblies to analyze, and additional bioinformatic skills are required to process the results and make them accessible in a user-friendly manner. Over the years, several tools and resources have been developed that use or integrate antiSMASH results, such as IMG-ABC [39] and MicroScope [40]. These do not specifically focus on marine species and provide results generated by previous versions of antiSMASH (v5 in IMG-ABC and v6 in MicroScope). The latest antiSMASH 7 detects more cluster types (81 versus 71 in version 6) and contains many improvements, such as showing all gene features in an interactive and filterable table for each region. Additionally, it provides quick options for exporting nucleotide or amino acid sequences for specific genes or regions as well as performing a direct BlastP search [15]. We developed the online platform MariClus, which can assist in identifying all putative gene clusters in a wide range of marine bacteria and archaea and, next to comparing the genome mining results to known clusters and products, also hint at new gene clusters for secondary metabolite biosynthesis with yet unknown functions. Our platform differs from existing ones by incorporating the latest antiSMASH 7 results from manually curated, high-quality genome assemblies, with a specific focus on marine species, by providing easy access to bioinformatic analyses for users without much bioinformatic expertise and by linking different public repositories to a central platform.

MariClus has several advantages over other existing resources for marine natural product discovery. Unlike MMP, which focuses on microbial genomics and metagenomics data [12], MariClus provides antiSMASH 7 results for biosynthetic gene clusters and predicted molecules from high-quality genome assemblies of marine bacteria and archaea. MMP does not offer any information on the biosynthetic potential or chemical diversity of the marine microbes in its databases. MariClus also differs from CMNPD, which is a knowledge base of chemical entities and biological activities of marine natural products [13]. CMNPD does not provide any genomic or biosynthetic information on the source organisms or the gene clusters responsible for the production of natural products. MariClus further differs from MarinLit, which is a database dedicated to marine natural products literature. MarinLit does not provide any genomic or biosynthetic information on the source organisms or the gene clusters either, and it is a subscription-based database [14]. It also does not use antiSMASH 7 results or MIBiG data for comparison. MariClus allows users to explore the link between gene clusters and molecules, as well as compare them to known clusters and products from MIBiG. MariClus also offers a user-friendly and intuitive web portal that allows users to search, filter and visualize the data on marine gene clusters and molecules. This platform goes beyond traditional marine investigations, as it enables users to discover various bacteria, gene clusters, environmental niches, metabolites and antibiotics.

In the future, MariClus will be annually updated based on active requests from the scientific community and the research groups involved in the development. Users can submit their requests for new or updated genome assemblies via the contact form on the MariClus website. By keeping track of new antiSMASH versions and updating the database accordingly, MariClus provides the most comprehensive and up-to-date information on marine biosynthetic gene clusters and molecules. MariClus can serve as a valuable resource for marine natural products discovery and inspire further research on the biosynthetic potential and chemical diversity of marine microorganisms.

## 4. Materials and Methods

High-quality genomes of marine species were collected from NCBI Genome (https://www.ncbi.nlm.nih.gov/genome/) and manually curated. When multiple genome assemblies were available for the species, the representative or reference genome (GCF_accession number) was used. If no representative genome was available, the assembly with the highest quality (GCA_accession number) was selected if no more than 30 scaffolds were present. Genomes with a less favorable assembly level (>30 scaffolds) were not included for further analysis. We collected the PubMed IDs (PMID) of relevant papers from PubMed and searched for species names in BacDive [17] to confirm their isolation sources and retrieve their culture collection numbers. Using the taxonomy ID, we retrieved the full taxonomic lineage with all taxa from the NCBI Taxonomy database.

A Nextflow pipeline was developed to analyze genomes of interest. The pipeline used the NCBI Datasets command line tool and genome accession numbers to download GenBank files for each genome. These files contained both genomic sequences and gene annotations. Next, antiSMASH 7 [15] was run on the GenBank files to identify biosynthetic gene clusters and predict their products. While we mostly relied on annotation information from the GenBank files, for some genomes that lacked this information, we used Prodigal as the gene finding tool (provided by antiSMASH 7). Additionally, several flags were used to enhance our analysis, including: --cb-general for general clusterblast search, --cb-knownclusters for known clusterblast search and --cb-subclusters for subclusterblast search.

A custom Python script was used to parse the data from the antiSMASH files. The information extracted from the antiSMASH output included the locus on which the cluster was located, the start and end positions of the cluster, the number of genes in the cluster, the predicted type of product of the cluster, the predicted products (known clusterblast) for the cluster and the anchor, which was used to directly connect the relevant antiSMASH page to MariClus. Finally, a second Python script generated database insert statements based on both the manually collected data and the data extracted from the antiSMASH output.

A MariaDB database is used to store data, with multiple tables for efficient retrieval of genome-related information, including species, genome assemblies, gene clusters, products and relevant publications. The web interface was written in Bootstrap 5 and included custom CSS and JavaScript. The interaction with the database was written in PHP using the mysqli library.

## Figures and Tables

**Figure 1 marinedrugs-21-00449-f001:**
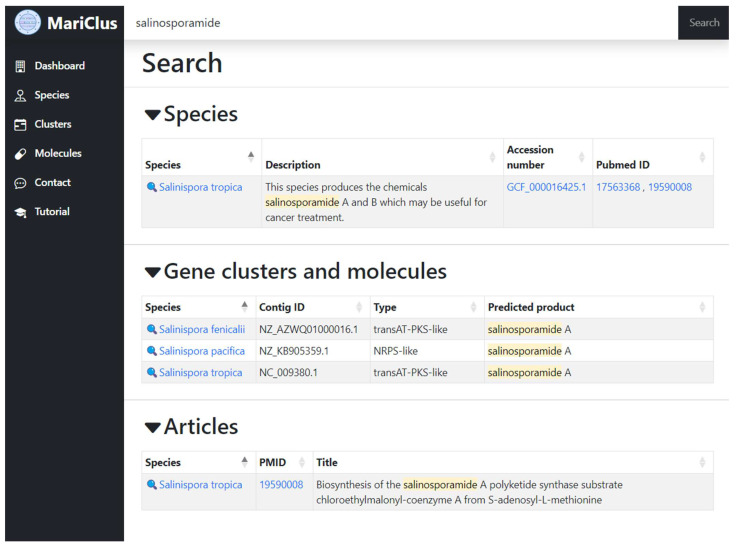
Screenshot of the MariClus database search function with the query ‘salinosporamide’. The search results are divided into three sections: species, clusters and molecules, and PubMed articles. The search term is highlighted in each result. The user can click on the icons or names to access more details about the species, clusters, molecules or articles.

**Figure 2 marinedrugs-21-00449-f002:**
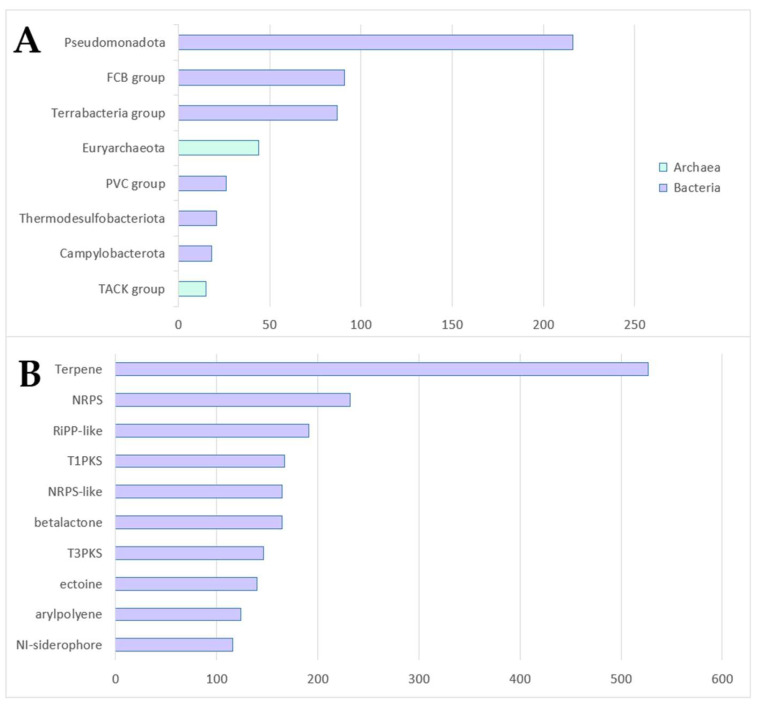
(**A**) The most common (super)phyla in the MariClus database. The colors indicate the superkingdom, as denoted by the legend. The most common type in the database is the Pseudomonadota group. This is also the largest group in the NCBI database. FCB group: Fibrobacterota, Chlorobiota and Bacteroidota group, PVC group: Planctomycetota, Verrucomicrobiota and Chlamydiota group, TACK group: Thaumarchaeota (now Nitrososphaerota), Aigarchaeota, Crenarchaeota (now Thermoproteota) and Korarchaeota group. (**B**) The most common cluster types predicted by antiSMASH 7 within the database. The terpene group is clearly the largest with over 500 clusters. NRPS: non-ribosomal peptide synthetase, RiPP: ribosomally synthesized and post-translationally modified peptide, T1PKS: type-1 polyketide synthase, T3PKS: type-3 polyketide synthase.

**Figure 3 marinedrugs-21-00449-f003:**
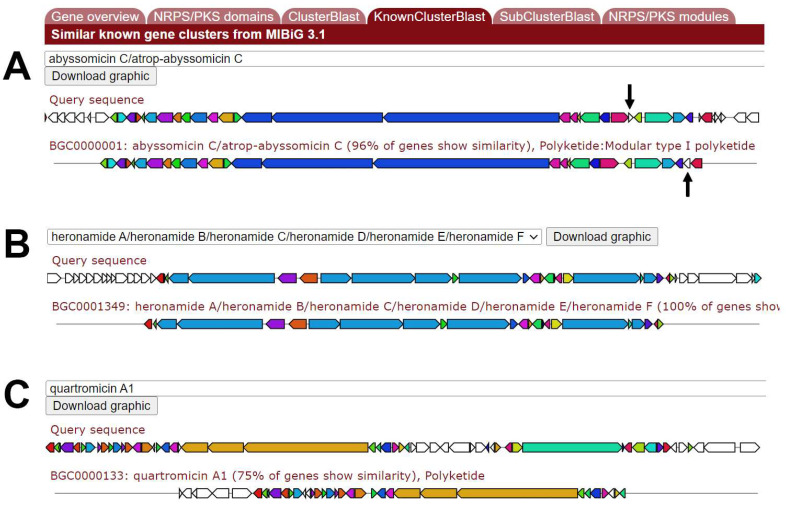
(**A**) Visualization of the BLAST results of the abyssomicin gene cluster in *Micromonospora maris*, identified via antiSMASH and generated with KnownClusterBlast. The query sequence on top represents the candidate gene cluster of *M. maris,* while the known abyssomicin C gene cluster of *M. maris* AB-18-032 [32] was derived from MIBiG and denoted as BGC0000001. Coloration of the coding sequences symbolizes the relationship between the genes of the known and candidate cluster. The one difference between both gene clusters is indicated by the arrow. (**B**) Visualization of the heronamide gene cluster in *Streptomyces marincola*. (**C**) Visualization of the quartromicin A1 gene cluster in *Amycolatopsis albispora*.

**Table 1 marinedrugs-21-00449-t001:** Overview on the top hits when searching for the abyssomycin gene cluster in MariClus. Candidates were ranked based on the cumulative BLAST score with the known abyssomicin C gene cluster of *Micromonospora maris* AB-18-032. The coverage represents the number of proteins with BLAST hits to this cluster out of the total number of genes in the cluster. This means that a coverage greater than 100% is possible if the number of unique query sequences with BLAST hits to the antiSMASH predicted cluster is larger than the total number of genes in the BGC cluster. Values of the cumulative BLAST score and coverage were obtained from antiSMASH.

Cumulative BLAST Score for …				
Species	… Abyssomicin	… Best Hit	Coverage Best Hit	Compound Prediction
*Micromonospora maris*	38,594	38,594	96%	abyssomicin C
*Streptomyces buecherae*	38,260	161,434	106%	nigericin
*Saccharomonospora piscinae*	31,152	241,330	56%	mediomycin A
*Streptomyces marincola*	25,243	120,655	100%	heronamide
*Actinoalloteichus fjordicus*	17,602	79,117	16%	mediomycin A
*Amycolatopsis albispora*	14,902	35,415	75%	quartromicin A1

## Data Availability

The data used for the MariClus database are publicly available in NCBI at https://www.ncbi.nlm.nih.gov/genome/browse#!/prokaryotes/ (access on 16 June 2023). The genome assemblies and antiSMASH results of the marine species can be accessed and downloaded through the MariClus web portal at https://mariclus.org/ (access on 6 July 2023).

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
