# Peer review of "MariClus: Your One-Stop Platform for Information on Marine Natural Products, Their Gene Clusters and Producing Organisms"

_marinedrugs, 2023, doi:10.3390/md21080449_

Round 1

Reviewer 1 Report

This manuscript outlines a new tool, MariClus, as a platform for mining and visualizing biosynthetic gene clusters (BGCs) from marine prokaryotes. As presented this tool is nicely laid out, easy to use (even for me as a non-computational biologist) and the data is delivered in a straightforward manner. The authors nicely describe the features of the tool, the content of the underlying database but also share the limitations of what can be generated. I appreciate the inclusion of the case studies that aid in demonstrating the usability of the tool. My only question is how extendable the tool is to other types of organisms e.g. algae, fungi, and can it be scaled as more marine sequences are generated?

Author Response

This manuscript outlines a new tool, MariClus, as a platform for mining and visualizing biosynthetic gene clusters (BGCs) from marine prokaryotes. As presented this tool is nicely laid out, easy to use (even for me as a non-computational biologist) and the data is delivered in a straightforward manner. The authors nicely describe the features of the tool, the content of the underlying database but also share the limitations of what can be generated. I appreciate the inclusion of the case studies that aid in demonstrating the usability of the tool. My only question is how extendable the tool is to other types of organisms e.g. algae, fungi, and can it be scaled as more marine sequences are generated?

We are glad to hear the reviewer’s positive feedback and their experience with the tool. Our aim is to provide a user-friendly platform for computational and non-computational scientists and are happy that the tool is straightforward. Besides maintaining the current version, our goal is to expand the platform over the coming years with additional tools and data. Extending the database to other marine organisms such as fungi, is one of our top priorities. We did not include it at this stage as we manually curate all the data ourselves, taking up a lot of time. Genome mining and gene cluster prediction is currently well-documented for prokaryotes and fungi through antiSMASH, however, no version specifically for algae has been developed. Should such tools become available in the future, the platform can be extended to algae or other marine species as well.

Reviewer 2 Report

The manuscript describes the new platform for the analysis of biosynthetic gene clusters in marine microorganisms. 

The text is well-written and organized. 

Regarding the database itself, it will be nice to include the option to select a data set based on Taxonomy (species, genus, family, etc. ). I could not find this option in the present form. 

I have a question, if authors consider of addition of marine fungal data to these database. 

Author Response

The manuscript describes the new platform for the analysis of biosynthetic gene clusters in marine microorganisms. The text is well-written and organized. Regarding the database itself, it will be nice to include the option to select a data set based on Taxonomy (species, genus, family, etc. ). I could not find this option in the present form.

We thank the reviewer for their suggestion to enable data analysis based on taxonomy. This option is currently not available and cannot be implemented as the taxonomy data is stored dynamically. To make this possible, we would have to make significant changes to the database structure, which is not feasible in the timeframe of these revisions. However, we value this suggestion and will keep it in mind for our next major database update. Finally, we would like to point out that at this stage, all taxonomy information directly links to NCBI Taxonomy, enabling taxonomical analysis.

I have a question, if authors consider of addition of marine fungal data to these database.

This is definitely ranked high on our future features list, as explained to reviewer 1.

Reviewer 3 Report

The authors report in this paper that they have constructed MariClus, a database of bacterial secondary metabolite synthesis clusters isolated in the ocean.

The oceans have potential to isolate many unknown bacteria and are a frontier as a genetic resource for secondary metabolite synthesis. Therefore, a database compiling BGCs of known marine bacteria will be an important research tool for researchers who share the same goal.

I hope that improvements can then be made in the following points.

In the information page for individual microorganisms, the BGC list is displayed at the bottom of the page, but the Predicted Product column is messy and difficult to understand. Terms such as "Cumulative BLAST Score" and "Known Cluster Coverage" are too long and their scores are difficult for a non-professional to understand. It would be good to improve the presentation to make it easier to understand, such as using abbreviations and coloring scores above a certain level.

In the case of Salinosporamide A, which the authors used as an example, it would be easier to understand if it is indicated on the page for known producing bacteria (Salinispora tropica), such as those registered in MiBIG. Currently that fact can only be inferred from the BLAST score and cluster gene list.

I followed hyperlink to the page with the results of the antiSMASH run, but only the cluster number is shown, and nowhere does it indicate what cluster number it was that I was trying to check.

The following is a minor point, 

I think it would be better to write a statement that would give some indication if it was a "No BGC detected".

The BGC type is underlined in the table, but I think it is better not to underline it because it can be misunderstood if there is a hyperlink.

Author Response

The authors report in this paper that they have constructed MariClus, a database of bacterial secondary metabolite synthesis clusters isolated in the ocean. The oceans have potential to isolate many unknown bacteria and are a frontier as a genetic resource for secondary metabolite synthesis. Therefore, a database compiling BGCs of known marine bacteria will be an important research tool for researchers who share the same goal. I hope that improvements can then be made in the following points.

In the information page for individual microorganisms, the BGC list is displayed at the bottom of the page, but the Predicted Product column is messy and difficult to understand. Terms such as "Cumulative BLAST Score" and "Known Cluster Coverage" are too long and their scores are difficult for a non-professional to understand. It would be good to improve the presentation to make it easier to understand, such as using abbreviations and coloring scores above a certain level.

We appreciate the reviewer’s remark and have made some quality of life modifications. First, we shortened the Predict Product column per reviewer’s suggestion by abbreviating the Cumulative BLAST Score to ‘Score’ and Known Cluster Coverage to ‘Coverage’. Additional information on the Score and Coverage is now given as a tool tip which is accessed by hovering the cursor over Score or Coverage. A second quality of life update was executed by ranking the predicted products of the same cluster in alphabetical order. For example, the prediction of arimetamycin A, B and C have the same cumulative BLAST score for various predicted gene clusters in Actinoalloteichus fjordicus, and the predicted products are now given in the order A, B and then C instead of A, C and then B. Finally, we’d also like to point out that reviewer 1 mentioned the platform is easy to use and because of this we do not wish to make too many changes. The use of coloring scores might be misleading to scientists outside the natural product discovery and genome mining field. By only providing the values, we stimulate the user to make their own analysis and interpretation of the data.

In the case of Salinosporamide A, which the authors used as an example, it would be easier to understand if it is indicated on the page for known producing bacteria (Salinispora tropica), such as those registered in MiBIG. Currently that fact can only be inferred from the BLAST score and cluster gene list.

At the moment, the origin of the known gene clusters from MIBiG used in product prediction is not highlighted and does indeed need to be inferred from the MIBiG database. A link to the exact page is provided on MariClus, however, addition of this data directly on MariClus warrants a substantial modification of the database, which is not feasible in this short timeframe. As mentioned with reviewers 1 and 2 pertaining the fungal data and taxonomy analysis, we aim to extend the platform with future features. Improved visibility on the known gene clusters could be part of a major database update in the near-future.

I followed hyperlink to the page with the results of the antiSMASH run, but only the cluster number is shown, and nowhere does it indicate what cluster number it was that I was trying to check.

When following the hyperlink to an antiSMASH page of a predicted gene cluster, the overview of all clusters of that genome assembly is loaded first, after which it will redirect to the corresponding gene cluster. The loading time is very short but should always redirect to the specific genomic region. Once the specific gene cluster prediction is shown, the region number is highlighted on the top (boxed region number). While we understand that the highlighting might be overlooked, it is not something we can stress more as it is part of the antiSMASH software.

The following is a minor point, I think it would be better to write a statement that would give some indication if it was a "No BGC detected".

We appreciate the reviewers suggestion and have implemented their remark. In case no gene cluster was identified, the Predicted Product column now shows ‘No BGC detected’.

The BGC type is underlined in the table, but I think it is better not to underline it because it can be misunderstood if there is a hyperlink.

We understand the reviewer’s concern but have chosen to keep the dashed underlining of the BGC type as it indicates a tool tip is present. On the tutorial page, we have elaborated on the differences between collapsed information, hyperlinks and tool tips. Specifically, the tutorial page now states:

Mariclus uses different styles of underlined text. An overview is given below:

  • Black underlined text means that the text can be clicked to expand or collapse additional information. It can also link to information on the same page (like the table of contents on this page)
  • Blue underlined text means that the text is a hyperlink to another (internal or external) page
  • Dashed lines means that the text contains additional information when hovering over it